# Rethinking end-to-end prediction of adsorption energies from a causal perspective

## Abstract

Adsorption energy is an important descriptor of catalytic activity in the field of catalysis, and significant efforts have been made to develop accurate predictive machine-learning models to replace expensive quantum chemistry calculations. Although it can be inferred by total energy predictions, research has mostly focused on the end-to-end prediction of adsorption energies due to the common belief that total energy is more challenging to predict than adsorption energy. In this study, we first analyzed the causal graph of adsorption energies and revealed that the indirect approach, which infers adsorption energy from total energy predictions, could provide better identifiability, leading to improved accuracy and generalization ability. We also improved the graph property normalization method for total energy prediction and achieved a halved Mean Absolute Error compared to direct adsorption energy prediction for the catalyst in-domain scenario. In the more challenging catalyst out-of-domain scenario, we found that the error primarily comes from predicting the individual energy of unseen catalyst atoms, and the error can be canceled when total energy predictions are used to infer adsorption energy. Consequently, our model achieves a MAE of approximately 0.2 eV for all tasks in the OC20 S2EF task, outperforming end-to-end models trained on datasets $50\times$ larger. Given the evidence presented in this study, future research should prioritize the development of total energy models to enhance the accuracy and efficiency of machine-learning approaches in material discovery.

## 1 Introduction

Catalysts play a crucial role in modern human society, for example, nearly 50% of the nitrogen found in human tissues originates from the Haber–Bosch process, which relies heavily on catalysts to enhance efficiency and reduce energy consumption, making it economically viable. However, discovering new catalysts is a lengthy process, typically taking one to two decades from discovery to commercial application (Fetanat et al., 2021). Most catalysts were historically discovered through trial and error, but modern advancements in quantum chemistry and computational chemistry have greatly accelerated the process. For example, the electron donor-acceptor theory (Tanaka et al., 1968) led to the development of a complex catalyst for the Haber-Bosch Process, consisting of alkali metal (electron donor), graphite (electron acceptor), and a transition metal compound, which demonstrates significantly better efficiency than traditional iron-based catalysts. This catalyst was further improved through quantum chemistry calculations, which identified the active sites of Ruthenium-based catalysts (Dahl et al., 1999; 2000) and led to the development of Barium-promoted oxide-supported Ruthenium (Bielawa et al., 2001), a catalyst now used in industry.

The binding process between catalysts and adsorbates is critical in catalytic chemical reactions. The process involves atoms moving to positions of lower energy, resulting in relaxed structures – the lowest energy configurations – and the corresponding relaxed energy. The physics of this binding process can be modelled by the causal graph in Figure 1, where the adsorption energy, defined as the difference in system energy before and after the binding process, is one of the most important descriptor for catalyst screening.

Quantum Chemistry calculation, especially Density Functional Theory (DFT), offers a more efficient alternative to real experiments in catalyst discovery (Nørskov et al., 2014; Zitnick et al., 2020). It can be used to infer the relaxed structure of catalyst, adsorbates, adsorbate-catalyst slab (denoted as

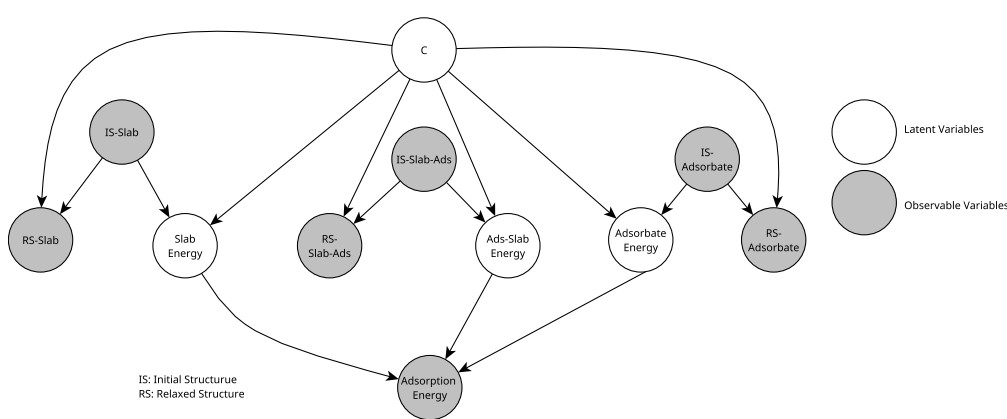

Figure 1: The causal model for the adsorption energy. The binding of a catalyst (usually referred as slab) and an adsorbate (molecule or its fragement) results in the enthalpy change, which can be reflected by the adsorption energy. The reaction enthalpy is the main contributor to the landscape. For a chemical reaction, the structure of chemicals can be determined using techniques such as X-ray diffraction (XRD) or Cryogenic electron microscopy (Cryo-EM), and the adsorption energy can be measured via the measurement of reaction enthalpy. Thus, these variables can be viewed as observable variables. Meanwhile, reaction conditions often cannot be fully obtained, as well as the total energy. Thus, they are considered latent variables.

adsorbate-slab for simplicity) bindings , as well as the corresponding energies, from which adsorption energy is computed. However, DFT requires a significant amount of computational resources, making high-throughput screening of catalysts, especially complex ones, computationally unfeasible. Recently, based on the great success of deep learning techniques, particularly the emergence of Graph Neural Networks (GNNs), a group of GNN based approaches has been applied to estimate the adsorption energies in an end-to-end manner - from the structure of adsorbate-slab bindings, *e.g.* SchNet (Schütt et al., 2017), DimeNet (Klicpera et al., 2020b;a), GemNet (Gasteiger et al., 2021), NequIP (Batzner et al., 2022), MACE (Batatia et al., 2022), PaiNN (Schütt et al., 2021), SCN (Zitnick et al., 2022), eSCN (Passaro & Zitnick, 2023), and Equiformer (Liao & Smidt, 2023; Liao et al., 2024). These GNN-based approaches achieve similar or even higher accuracy than DFT while requiring much less computational resources, thereby making high-throughput screening of catalyst computationally feasible. For example, it has been used to identify new alloys catalysts for carbon dioxide reduction and graphene production (Gu et al., 2020; Zhong et al., 2020; Li et al., 2024; 2023).

Despite significant progress, there is still an energy prediction gap that hinders the practical use of GNNs. Recently, Lan et al. (2023) show that current GNN-based approaches can predict relaxed structures well but fail to accurately predict adsorption energy. They showed that GNN-based approaches, combined with DFT single-point energy calculations, can achieve an accuracy of approximately 90% in finding the global minima adsorption energy, close to the full DFT approach (DFT relaxation and DFT single-point calculation), while the accuracy of GNN-only approaches drops to approximately 50%. Although previously empirical results suggest that the end-to-end prediction framework from structure to adsorption energies outperforms other approaches, we argue that the end-to-end framework is a major source of poor energy estimation accuracy.

The causal graph shown in Figure 1, which aligns with the physical process of catalyst-adsorbate binding. In real-world chemical systems, the relaxation process occurs when the material is subjected to several latent factors, such as reaction conditions, allowing atoms to gradually shift from initial positions to positions of lower energy. In this process, the energy of catalyst, adsorbate and adsorbate-slab binding are latent variables since they cannot be measured by experiments. Meanwhile, the adsorption energy is an observable variable as it can be obtained by measuring the reaction enthalpy. Similarly, relaxed structures can also be measured by techniques such as X-ray diffraction (XRD) or Cryogenic electron microscopy (Cryo-EM), making them observable variables.

The end-to-end prediction framework requires identifing latent factors from the initial structure of adsorbate-slab bindings, closely related to latent causal representation learning. Very recently, progress has been made in the theory of identifiability of latent causal representation learning. Typically, to identify latent causal variables/confounders, it is required that there should be a one-to-one mapping from observations to latent factors, up to some random noise (Zhang et al., 2024; Zhou et al., 2023; Liu et al., 2022; Von Kügelgen et al., 2021). Failure to identify latent causal factors or fit the latent causal model usually results in poor generalization ability and out-of-domain performance (Richens & Everitt, 2024). Meanwhile, out-of-domain performance is crucial in catalyst screening, as good catalysts often have some unique properties not demonstrated by other materials.

Non-end-to-end approaches require total energy models to recover latent variables – the energy of adsorbate-slab, slab and adsorbate. Recently, Tran et al. (2023) explored the task of total energy prediction from the adsorbate-slab binding strucutures and reported that it often shows worse accuracy than adsorption energy prediction, particularly in out-of-domain situations. This result suggests that predicting total energy is a more challenging task than predicting adsorption energy, and end-to-end adsorption prediction is preferred. However, we argue that the total energy of slab and adsorbate-slab structure, which serves as latent variables in the causal graph, can often be recovered to some equivalent class. Particularly for the catalyst slab and the adsorbate-slab structure, if their total energies are consistently higher (or lower) than the ground truth, the energy profile of the reaction would not be affected, nor would the adsorption energy. For adsorbate-slab binding process, as the energy of adsorbate can be considered known, we only need to predict the total energy of catalyst slab and the adsorbate-slab binding structure. Since the relaxed slab and the adsorbate-slab complex have similar structure in most cases, it is natural to assume that the model would have similar biases in their energy predictions, and the error induced by the epistemic uncertainties would be canceled in the final inferred adsorption energy.

Achieving error cancellation for inferred adsorption energy requires highly accurate total energy prediction; otherwise, errors may even accumulate. We enhanced the accuracy of total energy prediction by improving graph property normalization. This technique assigns each node a coefficient and subtracts the target by the sum of these node coefficients, effectively reducing aleatoric uncertainty by decreasing the variance in data caused by variations in graph sizes or compositions, and allows the model to focus on structural features rather than node count. Notably, this approach was recognized in some of the earliest machine learning studies in this field (Rupp et al., 2012; Montavon et al., 2012; Hansen et al., 2013), which demonstrated that predicting atomization energy – a equivalent form of graph property normalization – is easier than predicting raw total energy. However, in the context of predicting catalyst-related energy, normalization has been largely overlooked by the community, potentially because the shift in focus from total energy to adsorption energy, as the later one does not require graph property normalization.

In this study, we demonstrate that the indirect approach, which infers adsorption energy from total energies, can provide higher accuracy and better generalization. We also designed and compared several graph normalization methods and found that per-element normalization yields the highest accuracy, achieving halved MAEs compared to direct end-to-end adsorption energy prediction for in-domain (ID) and out-of-domain adsorbate (OOD Ads) cases. For out-of-domain catalyst (OOD Cat) and out-of-domain both (OOD Both) cases, the prediction error remains higher than the direct approach. However, the biases are consistent for both adsorbate-slab and slab in these two cases, thus canceling each other out when used to infer adsorption energy. This leads to a 41% improvement in the MAE for OOD Ads and OOD Both. This highlights the potential of using total energy predictions to infer adsorption energies, offering a promising pathway to more accurate catalyst screening. With the results presented in this work, we suggest that the community focus on developing total energy predictions to accelerate the practical use of ML catalyst discovery models.

## 2 RELATED WORK

### 2.1 REPRESENTATION-BASED POTENTIAL AND GRAPH NEURAL NETOWRK POTENTIAL

The primary challenge in developing machine learning potentials is accurately predicting the forces and energies associated with a given set of atoms and their positions. Prior to the emerging of deep neural networks, research mostly focused on designing representations that describe the atomic environment and fitting these representations to target forces or energies. Notable efforts in this

direction include Couloumb Matrices (Rupp et al., 2012; Montavon et al., 2012), Bags of Bonds (Hansen et al., 2015), Atom-centered Symmetry Functions (Behler, 2011; Smith et al., 2017a;b), Smooth Overlap of Atomic Positions (Bartók et al., 2017) and FCHL19 (Christensen et al., 2020). However, these representations face limitations, such as restricted scalability and inadequate capture of long-range interactions.

To achieve scalability and end-to-end learning, researchers shifted their attention to Graph Neural Networks (GNNs). GNNs leverage graph mathematical formulations to model molecular properties by representing molecules as graphs, where atoms are nodes and edges are defined by inter-atomic distances (or bonds). Let $G = (V, E)$ denote a molecular graph, where $V$ is the set of atoms and $E \subseteq V \times V$ is the set of edges. Each atom $v_i \in V$ is associated with a feature vector $x_i$ representing its chemical properties, and each edge $e_{ij} \in E$ is characterized by a feature vector $e_{ij}$ encoding the distance between atoms $v_i$ and $v_j$. GNNs employ message-passing algorithms (Gilmer et al., 2017) to propagate information between neighbouring nodes, iteratively updating node features based on their local graph neighbourhoods. To enhance expressiveness, two approaches – higher-order invariant models and equivariant models are employed.

Early works on GNN potentials used the molecular graph definition where atoms are nodes and bonds (or atomic distances) are edges. It was found that GNN based on this kind of graph has weak expressiveness. To address this, higher order invariant features such as bond angle and dihedral angle were introduced in models such as DimeNet and GemNet (Klicpera et al., 2020b; Gasteiger et al., 2021), enhancing their expressiveness and sensitivity to geometric changes. However, the computational cost is substantial, so in practice, GemNet (Gasteiger et al., 2021) calculates dihedral angles only in critical regions, such as the adsorption for adsorbate-slab structures.

Other than invariant higher order features, Equivariant Graph Neural Networks (EGNNs) address this limitation by incorporating rotation invariance and equivariance directly into their architecture by using Group Convolution (Cohen & Welling, 2016), Spherical Convolutions (Cohen et al., 2018), Equivariant message passing (Batzner et al., 2022; Yan et al., 2024), Equivariant Attention (Hutchinson et al., 2021; Frank et al., 2022; Liao & Smidt, 2023).

## 2.2 Graph Property Normalization

Graph Property Normalization, which transfers the original GNN target $y_{DFT}$ into a normalized target $y_{stand}$, is an effective way to improve the accuracy of ML models. In this specific application, it can be understood as substituting the DFT energy by the sum of per-element energies, and can be classified as atomization energy normalization (Rupp et al., 2012; Montavon et al., 2012; Hansen et al., 2013) and statistical normalization (Tran et al., 2023). The definition of atomization energy - the energy change after a molecule is separated into individual atoms, suggests that the atomization energy is a physical approach. It calculates the energy of each elemental single atom in the gas-phase using DFT, and then get the normalized target using the formula of

$$y_{stand} = y_{DFT} - \langle \mathbf{k}, \mathbf{p} \rangle \tag{1}$$

where $y_{DFT}$ is the original DFT energy, $\mathbf{k}$ is a counting vector which counts the number of a specific element, and $\mathbf{k}$ is the calculated per-element single atom energy vector.

In contrast, Tran et al. (2023) proposed the statistical approach which using ordinary least square (OLS) equations to get the coefficients vector $\mathbf{p}$ by minimize $y_{DFT} - \langle \mathbf{k}, \mathbf{p} \rangle$, as

$$\hat{\mathbf{p}} = \arg \min_{\mathbf{p}} \left\{ \| y_{DFT} - \langle \mathbf{k}, \mathbf{p} \rangle \|_2^2 \right\} \tag{2}$$

However, the use of OLS may leads to the coefficient to be sensitive to outliers and resulted in false coefficients.

## 3 Proposed Method

### 3.1 Infer Adsorption Energy Indirectly by Total Energy Predictions

Figure 1 illustrates the general workflow for calculating adsorption energy in computational chemistry. The process begins by constructing a pure surface slab, which undergoes DFT relaxation to

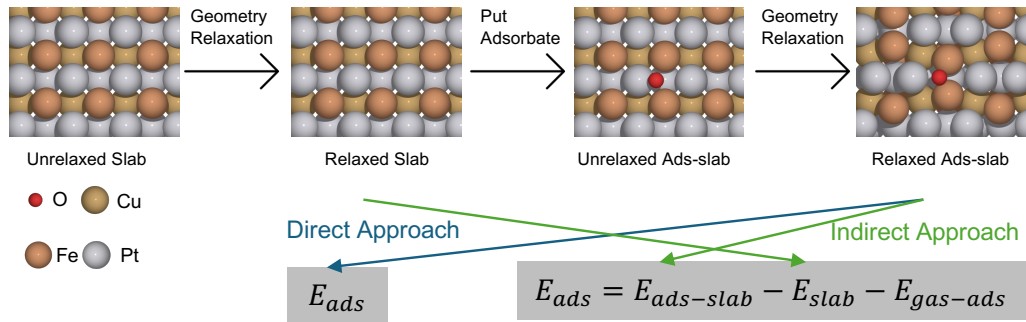

Figure 2: Schematic of the workflow used to calculate adsorption energy in computational chemistry. The process begins with the calculation of the relaxed slab energy, followed by the calculation of the relaxed adsorbate-slab energy to derive the adsorption energy. A significant geometry change is observed after the adsorbate-slab geometry relaxation, where the iron (Fe) atom is pulled by the oxygen, and changed their positions. The two equations summarize the corresponding inputs for direct and indirect approaches.

yield a relaxed pure slab. Next, an adsorbate is placed on the surface, and DFT relaxation is performed again to obtain a relaxed adsorbate-slab (ads-slab) structure. The adsorption energy is then calculated using the equation $E_{\text{ads}} = E_{\text{ads-slab}} - E_{\text{slab}} - E_{\text{gas-ads}}$, which corresponds to the total energy of adsorbate-slab, pure slab and gas-phase adsorbate structure, respectively. The labeling process of OC20 data set follows the same process.

According to the equation, $E_{\text{ads}} = E_{\text{ads-slab}} - E_{\text{slab}} - E_{\text{gas-ads}}$, the necessary information needed for $E_{\text{ads}}$ should be the structures that correspond the later three terms. However, the direct approach only uses relaxed adsorbate-slab structure to predict $E_{\text{ads}}$, therefore assuming that both $E_{\text{ads-slab}}$ and $E_{\text{slab}}$ can be learned implicitly. This assumption, to some extent, is a good one since the pure slab and the slab under adsorption conditions normally only have small geometry differences.

However, the geometry difference is small is just a general observation, and in cases where there is a big geometry difference, for example, in Figure 2, the two iron atoms have been strongly pulled by the oxygen adsorbate, we can expect some performance decrease for this approach. Besides, this assumption also explains why all state-of-the-art models reaches a success rate bottleneck in finding the global-minima adsorption energy, despite the fact some of them obtained lower prediction error. To mitigate the information loss, a more effective method should be predicting total energies using the corresponding structures, and infer adsorption energies from there.

**Overall Framework**   We propose using the equation for calculating adsorption energy to predict adsorption energy. According to this equation, it requires the prediction of $E_{\text{ads-slab}}$, $E_{\text{slab}}$, and $E_{\text{gas-ads}}$. Since $E_{\text{gas-ads}}$ is typically a constant, it can be disregarded, leaving $E_{\text{ads-slab}}$ and $E_{\text{slab}}$ as the primary variables to be predicted. For both of these variables, we employed normalization by regularized per-element coefficients to enhance their prediction accuracy. The improved predictions can then be used to infer the adsorption energy.

**Identifiability of latent variables**   Theoretically, considering the causal graph in Figure 1, even without knowing the structures of slab, it may still be possible for a model to infer the total energy of the pure slab given several additional information. For example, by slightly extending the theoretical framework in Liu et al. (2024), one can show that if there is an invariant and invertible mapping from the adsorbate-slab structure to the relaxed slab structure, it may be possible to recover the latent energy of the slab up to some equivalent class. However, the assumption of invariance and invertibility may be unrealistic in our application. Theoretically, since the relaxation procedure of different chemicals are driven by the same mechanism, there may exist an invariant mapping from the relaxed adsorbate-slab structure to the relaxed slab structure. However, the factors influencing the reaction are not yet fully known, and not all of them can be observed. As a result, in practice, for the binding process of the catalyst and adsorbate, an invariant mapping from the relaxed adsorbate-slab structure to the relaxed slab structure is not observed. This is supported by the fact

the relaxation-based approaches outperform direct approaches for the Initial Structure to Relaxed Energy (IS2RE) task in the Open Catalyst Challenge (Chanussot et al., 2021).

For end-to-end adsorption energy prediction models, it actually assumes that there exists invariant mapping from relaxed adsorbate-slab structure to the relaxed slab structure, and this invariant mapping is implicitly learned by the GNNs. Therefore, it would be natural to predict that the performance of these models would be good for ID cases, but poor for OOD catalysts and adsorbates.

**Regularized Graph Property Normalization**   Different with Tran et al. (2023) which calculates the $\mathbf{p}$ vector using the OLS equation, we employed a ridge regression, as

$$\hat{\mathbf{p}} = \arg\min_{\mathbf{p}} \left\{ \|y_{DFT} - \langle \mathbf{k}, \mathbf{p} \rangle\|_2^2 + \lambda \|\mathbf{p}\|_2^2 \right\} \tag{3}$$

where $\lambda$ is the regularization parameter, and a fixed value of 0.001 was used in this study. Figure S1 in the Supplementary Material demonstrate how this method effectively mitigates the influence of outliers and reduces the std of normalized target. After graph normalization for each sample in the dataset, the normalized energy $y_{DFT} - \langle \mathbf{k}, \mathbf{p} \rangle$ serves as the training target for graph neural networks.

Normalization by per-element coefficients can overlook elements not present in the training set, leading to significant performance drop if out-of-domain elements are included in the test test. To address this, we proposed Normalization by per-Group coefficient (GN) and Normalization by per-atom Uniform coefficient (UN). Both GN and UN use the same equation as Equation (2) and Equation (3), but the $\mathbf{k}$ would count the number of the same group elements and number of atoms, respectively. Therefore, GN generate a per-group coefficient and UN generate a per-atom coefficient. Both method have better generalization than EN since they can handle cases that the test elements do not appear in the training set.

## 4   RESULTS

### 4.1   INFER ADSORPTION ENERGY INDIRECTLY FROM TOTAL ENERGY

We begin our discussion with an empirical evaluation of the direct approach and indirect approach. Equiformer V2 (denoted as EqV2) (Liao et al., 2024) and eSCN (Passaro & Zitnick, 2023) are selected as the GNN architectures for the indirect approach. Both of them, along with two additional architectures (SCN (Zitnick et al., 2022) and GemNet-OC (Gasteiger et al., 2021)), are also selected for the direct approach using their publicly available weights. Details of the training process and the public checkpoints used can be found in the Supplementary Material.

Table 1 compares the MAEs in predicting adsorption energy using two approaches, as well as the individual MAEs in predicting adsorbate-slab and slab total energies. The indirect-EqV2 achieved the highest accuracy with MAEs lower than 0.2 for all splits, except for OOD Both. In comparison to direct-EqV2, the improvement is particularly significant for OOD Ads and OOD Both, which achieve MAEs that are 59% of direct prediction values. The MAEs of the total energies could provide further information – the EN-EqV2 obtained MAEs of 0.106 and 0.139 for ID and OOD Ads adsorbate-slab, much lower than the direct approach. However, the MAEs for the slab are higher (0.208 for ID and 0.219 for OOD Ads), which means the accuracy of the indirect approach is mainly affected by this. The EN-EqV2 obtained much higher MAEs in predicting adsorbate-slab total energy for OOD Cat and OOD Both, and even worsened MAEs for pure slab (e.g. for OOD Cat, 0.314 for adsorbate-slab and 0.388 for pure slab); however, when these values are used to infer adsorption energy, the errors are canceled out and the MAE is reduced to 0.191 and 0.216. The same phenomenon is observed for eSCN as well - EN-eSCN obtained high MAEs for OOD Cat and OOD Both, for both adsorbate-slab and slab total energies, but the errors are canceled out when they are used to infer adsorption energy, resulting in higher accuracy than the direct approach. Figure S3 plots the total energy error of the adsorbate-slab system against that of the pure slab. The plot reveals a strong correlation between them in OOD Cat and OOD Both cases, and providing an explanation for the occurrence of error cancellation. Therefore, we can make the observation that the error in predicting total energies of catalyst OOD scenarios primarily comes from the error in predicting the atoms that the model has not encountered before, and this error can be canceled because they will bias in the same direction.

Table 1: MAE of Different Methods for the S2EF task. Indirect-EqV2 consistently outperforms other methods, achieving the lowest MAE across all categories. Direct approaches generally have higher errors in OOD Ads and OOD Both. For total energy predictions, EN-EqV2 shows a higher MAE for slab energies compared to adsorbate-slab energies. This discrepancy leads to higher inferred adsorption energy errors in ID and OOD Ads cases, but lower errors for OOD Cat and OOD Both due to error cancellation

| Method | Dataset | Target | ID | OOD Ads | OOD Cat | OOD Both |
|---|---|---|---|---|---|---|
| Indirect-EqV2 | OC20-2M | Adsorption | **0.180** | **0.183** | **0.191** | **0.216** |
| Indirect-eSCN | OC20-2M | Adsorption | 0.264 | 0.260 | 0.256 | 0.262 |
| Direct-EqV2 | OC20-2M | Adsorption | 0.283 | 0.319 | 0.271 | 0.365 |
| Direct-eSCN | OC20-2M | Adsorption | 0.294 | 0.312 | 0.288 | 0.378 |
| Direct- SCN | OC20-2M | Adsorption | 0.313 | 0.329 | 0.297 | 0.380 |
| Direct-GemNet-OC | OC20-2M | Adsorption | 0.297 | 0.314 | 0.296 | 0.372 |
| EN-EqV2 | OC20-2M | Ads-Slab Total | 0.107 | 0.139 | 0.314 | 0.343 |
| EN-EqV2 | OC20-2M | Slab Total | 0.208 | 0.219 | 0.388 | 0.429 |
| EN-eSCN | OC20-2M | Ads-Slab Total | 0.121 | 0.141 | 0.326 | 0.354 |
| EN-eSCN | OC20-2M | Slab Total | 0.272 | 0.276 | 0.423 | 0.454 |

*Note: For a fair comparison, this table includes only the samples for which we could identify the DFT slab structure from the OC20 "validation" set. This resulted in final sample counts of 540,062 for ID, 537,192 for OOD Ads, 999,809 for OOD Cat, and 999,804 for OOD Both. A comparison of Direct EqV2 and EN-EqV2 predictions without exclusions can be found in Table S2 in the Supplementary Material.*

Figure 3 compares the error distribution of the direct and indirect approaches using the best performing model EqV2. For the ID and OOD Cat (Figure 3 a, c), the direct approach exhibits a shallow peak, but the peak is much wider for OOD Ads and OOD Both, indicating that OOD Ads is challenging for it. Conversely, the indirect approach enhances performance primarily by reducing the number of samples in the long tail – samples with $|\text{Error}| > 0.5$. This observation is further supported by Figure S2, which demonstrates that unlike the direct approach, the indirect approach maintains similar error distributions across the four splits, indicating great generalization ability.

The high MAEs in the total energy of slabs are the primary source of errors for the indirect approach. (Tran et al., 2023) reported the high error for slab energy as well, showing that the error of pure slab is $1.5\times$ higher than adsorbate-slab, likely due to the much lower number of samples on the pure slab. However, there is another contributing factor in our study, which can potentially be mitigated. The OC20 2M dataset is supposed to contain 247,132 unique pure slab trajectories, but we were only able to access 178,763 slabs from the public data source. The remaining slabs are hidden to prevent test data leakage. Consequently, the total energy model used in this study was trained on a dataset with reduced sampling coverage. It is reasonable to assert that if the training set encompassed all slabs, the MAEs for the slab total energy and indirect adsorption energy inference could be further reduced. Additionally, since all slab trajectories for the test set are hidden, the results in Table 1 are based on the OC20 "validation" set rather than the test set. This is because the relaxed slab structure is an additional input for the indirect approach. It is important to note that the "validation" datasets are named as such following (Chanussot et al., 2021), but we only used a subset of 10,000 samples from Val ID for validation during training, and the results for them are excluded. Consequently, the ID, OOD Ads, OOD Cat, and OOD Both validation sets presented in Table 1 follow a rigorous testing procedure where the model had not encountered them before testing.

## 4.2 ABLATION STUDY

## 4.3 GRAPH PROPERTY NORMALIZATION IN PREDICTING TOTAL ENERGIES

The indirect approach has been demonstrated to achieve significantly higher accuracy in predicting adsorption energy. We now examine the contribution of each component to the overall performance. Initially, we compare the accuracy achieved using various graph property normalization methods. Table 2 presents the MAEs in predicting the total energy using different methods. Among all methods, the EN method achieved the lowest error across all splits. Compared to the direct approach, the MAEs for the ID and OOD Ads cases are nearly halved (0.112 vs. 0.213 for ID and 0.115 vs. 0.232 for OOD Ads). These values are even lower than those obtained by the direct approach trained

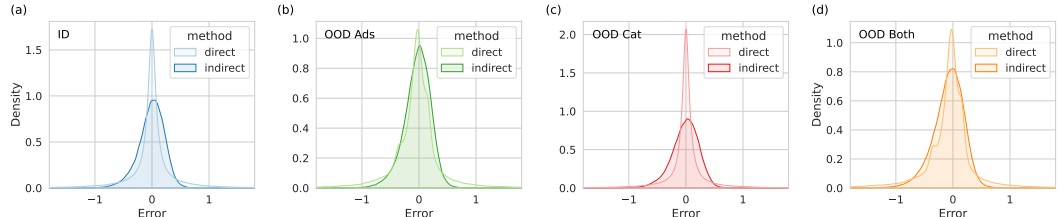

Figure 3: Error distribution using direct and indirect approaches for (a) ID, (b) OOD Ads, (c) OOD Cat and (d) OOD Both data sets. The direct approach exhibits a narrower distribution around zero for ID and OOD Cat, but a wider distribution for OOD Ads and OOD Both. The indirect approach demonstrates its effectiveness particularly by reducing the number of high-error samples (long-tail samples), especially in the cases of OOD Ads and OOD Cat, which have significantly fewer samples with $|\text{Error}| > 0.5$.

with the ALL+MD dataset, which is $50\times$ larger. However, for the OOD Cat and OOD Both cases, the error of the EN method is higher than that of the direct approach, indicating a challenge in the catalyst out-of-domain scenario. As shown in Table 1, this error can be mitigated when used to infer adsorption energy. Nevertheless, if the coefficients of the EN method are fitted using another dataset (OC22), the MAEs dramatically increase to the level of no-normalization (NN-EqV2), potentially due to some elements in OC20 not being included in the OC22 dataset. Apart from EN, the GN method ranks second among all normalization methods, followed by UN. UN still performs much better than no normalization, emphasizing the importance of graph property normalization for total energy prediction.

Figure 4 illustrates the normalized target as a function of the number of atoms using various methods, and partially explains the accuracy of different normalization techniques. The target, which is the total energy, is most effectively narrowed using the EN method, resulting in a standard deviation (std) of 19.62. This significantly reduces aleatoric uncertainty. In contrast, when coefficients are fitted for each group or uniformly, the standard deviations are considerably higher, at 66.08 and 151.39, respectively. The MAEs follow a similar order to the stds for these normalization methods. However, when the EN method was fitted by a different dataset (OC22), the resulting standard deviation is much higher at 155.718, which is approximately eight times larger than the coefficients fitted by the OC20 dataset.

The results from the OC22 EN method suggest a coefficient shifting issue, as the coefficients fitted by OC22 have no effect on OC20. This raises several questions: Is there a normalization method that can achieve the highest accuracy for both the OC20 and OC22 datasets? If so, would it be a statistical method, like the EN method used in this study, or a physical method, such as atomization energy? Due to the high computational resources required to train OC22 models, we leave these questions for future exploration.

Table 2: MAE in predicting the total energies for the S2EF test sets using different graph normalization methods. The EN method, which calculates per-element coefficients, performs the best, followed by normalization by per-group coefficients (GN) and normalization by per-atom uniform coefficients (UN). All normalization methods outperform the non-normalization, except for OC22-EN, which fits the coefficients from the OC22 dataset. Compared to the direct approach, EN-EqV2 achieves halved MAEs for ID and OOD Ads, but higher MAEs for OOD Cat and OOD Both.

| Method | Dataset | Target | ID | OOD Ads | OOD Cat | OOD Both |
|---|---|---|---|---|---|---|
| EN-EqV2 | OC20-2M | Total | **0.112** | **0.115** | 0.366 | 0.396 |
| GN-EqV2 | OC20-2M | Total | 0.145 | 0.147 | 0.417 | 0.460 |
| UN-EqV2 | OC20-2M | Total | 0.211 | 0.211 | 0.507 | 0.546 |
| NN-EqV2 | OC20-2M | Total | 0.337 | 0.332 | 0.590 | 0.687 |
| OC22-EN-EqV2 | OC20-2M | Total | 0.353 | 0.363 | 0.738 | 0.824 |
| Direct-EqV2 | OC20-2M | Adsorption | 0.213 | 0.232 | **0.263** | **0.349** |
| Direct-EqV2 | OC20 All+MD | Adsorption | 0.157 | 0.165 | 0.243 | 0.310 |

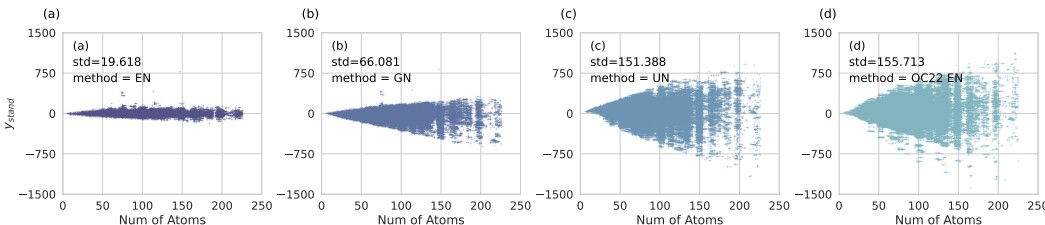

Figure 4: Normalized target values plotted against the number of atoms using normalization methods: (a) normalization by per-element coefficient (EN), (b) normalization by per-group coefficient (GN), (c) normalization by per-atom uniform coefficient (UN), and (d) EN fitted using the OC22 dataset. The standard deviations of the targets follow the order EN < GN < UN, which corresponds to the order of the achieved MAEs using these methods.

## 5 CONCLUSION

This work reevaluates the use of a direct end-to-end approach for predicting adsorption energy, analyzes the causal graph of adsorption energy, and proposes an indirect approach which infers adsorption energy from adsorbate-slab and pure slab total energy predictions. This indirect method offers better identifiability and generalization ability. We propose and systematically compare several graph property normalization methods, finding that per-element coefficients are most effective in reducing the standard deviation of total energy and improving performance. This approach significantly halves the MAEs in catalyst in-domain scenarios compared to the direct approach. For the more challenging out-of-domain catalyst scenarios, our results reveal that the higher error primarily arises from predicting the energy of previously unencountered slab atoms. This error can be mitigated when total energies are used to infer adsorption energy. Using our method, we achieved MAEs of approximately 0.2 for all four S2EF splits (ID, OOD Ads, OOD Cat, OOD Both). Additionally, we identified limitations in predicting pure slab energy due to insufficient sampling coverage. Addressing this issue by expanding the dataset with more slab samples could further improve the accuracy of the indirect approach. In conclusion, this study suggests a shift in focus towards developing total energy prediction models for more accurate adsorption energy prediction. This approach promises to enhance the practical use of machine learning models in catalytic material discovery, thereby contributing to the advancement of sustainable energy technologies.

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

# Appendix

## A  TRAINING AND EVALUATION DETAILS

In this study, we employ the implementation provided in the Open Catalyst Project (OCP) repository (`https://github.com/FAIR-Chem/fairchem`) for all models utilized. The OCP framework offers robust tools for training and evaluating machine learning models for catalytic systems, ensuring consistency and reproducibility across experiments.

For the training of both the Equviformer V2 and eSCN total energy models, we initialize the models using their pretrained 2M checkpoints (`https://fair-chem.github.io/core/model_checkpoints.html`) and finetune the model using a combined dataset. The combined dataset includes both the 2M dataset and a slab dataset (559,927 samples) randomly choose from the slab trajectories. The optimization is guided by a loss function which balance the error of energy and force prediction.

$$\mathcal{L}_{total} = \gamma_e \mathcal{L}_{energy} + \gamma_f \mathcal{L}_{force} \qquad (4)$$

where $\mathcal{L}_{energy}$ and $\mathcal{L}_{force}$ are the energy and force loss; $\gamma_e$ and $\gamma_f$ are the coefficients for energy and force loss, respectively. In this study, fixed values as $\gamma_e = 2$ and $\gamma_f = 100$ are used.

Additionally, all hyperparameters, optimization settings, and other key training configurations used in our study are summarized in the following table for reference and reproducibility.

Table 3: Hyperparameters used for EqV2 and eSCN total energy models

| Model | Hyperparameter | Value |
|-------|----------------|-------|
| EqV2 | learning rate | 5e-5 |
|      | num epochs | 20 |
|      | optimizer | AdamW |
|      | weight decay | 1e-3 |
| eSCN | learning rate | 5e-5 |
|      | num epochs | 15 |
|      | optimizer | AdamW |
|      | weight decay | 0 |

To evaluate the indirect approach, we compared Equviformer V2, eSCN, SCN, and GemNet-OC. We utilized the publicly available checkpoints for them, each trained on the 2M dataset. Specifically, the official checkpoints used can be found at `https://fair-chem.github.io/core/model_checkpoints.html` using the following identifier.:

Equviformer V2: EquiformerV2-83M-S2EF-OC20-2M

eSCN: eSCN-L4-M2-Lay12-S2EF-OC20-2M

SCN: SCN-S2EF-OC20-2M

GemNet-OC: GemNet-OC-S2EF-OC20-2M

## B  GNN FOR THE IS2RE TASK

We also evaluate indirect approach and total energy models in the Initial Structure to Relaxed Energy (IS2RE) task. Table S1 presents the metrics for the IS2RE test sets using the EN-EqV2 and Direct-EqV2. For total energy, similar to the results in Table 1 and Table 2, EN-EqV2 achieved lower MAEs than Direct-EqV2 for the ID and OOD Ads cases, but higher MAEs for the OOD Cat and OOD Both cases.

When using total energy to infer adsorption energy, error cancellation is again observed for the OOD Cat and OOD Both cases. However, all MAEs of the indirect approach are higher than direct approach. This is because the accuracy of indirect approach closely related to the accuracy of adsorbate-slab and pure slab total energy predictions. As discussed in the main text, the EN-EqV2

model trained in this study shows a limitation in predicting slab energy due to an insufficient number of samples and sampling coverage. Once the issue of slab samples and coverage are addressed, it is reasonable to expect that indirect approach will achieve lower MAE for the IS2RE task.

Table S1: MAE of Different Models for the IS2RE task. EN-EqV2 obtained lower MAE than direct-EqV2 for ID and OOD Ads, but when the EN-EqV2 predicted total energies being used to infer adsorption energy, all MAEs are higher than direct EqV2 due to the limitation of EN-EqV2 in predicting slab energies.

| Method | Dataset | Target | ID | OOD Ads | OOB Cat | OOD Both |
|---|---|---|---|---|---|---|
| Indirect-EqV2 | OC20-2M | Adsorption | 0.427 | 0.434 | 0.425 | 0.413 |
| Direct-EqV2 | OC20-2M | Adsorption | 0.345 | 0.375 | **0.353** | **0.333** |
| EN-EqV2 | OC20-2M | Total | **0.317** | **0.325** | 0.523 | 0.511 |
| Direct-EqV2 | OC20 All+MD | Adsorption | 0.301 | 0.317 | 0.331 | 0.290 |

Table S2: MAE for prediction adsortpion energies using OC20 S2EF validation dataset. Here the EqV2-Direct means predict adsorption energies directly, and DS-EqV2-Total means predict adsorption energies by adsorbate-slab total energy minus slab total energy

| Method | Dataset | Target | ID | OOD Ads | OOB Cat | OOD Both |
|---|---|---|---|---|---|---|
| Direct-EqV2 | OC20-2M | Adsorption | **0.217** | **0.261** | 0.271 | 0.365 |
| Direct-eSCN | OC20-2M | Adsorption | 0.227 | 0.254 | 0.288 | 0.378 |
| Direct- SCN | OC20-2M | Adsorption | 0.246 | 0.269 | 0.297 | 0.380 |
| Direct-GemNet-OC | OC20-2M | Adsorption | 0.231 | 0.254 | 0.296 | 0.372 |
| EN-EqV2 | OC20-2M | Ads-Slab Total | 0.093 | 0.118 | 0.314 | 0.343 |
| EN-eSCN | OC20-2M | Ads-Slab Total | 0.104 | 0.122 | 0.326 | 0.354 |

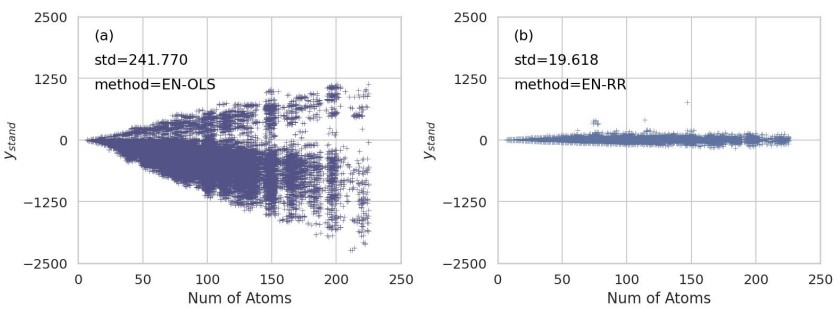

Figure S1: Normalized target values plotted against the number of atoms using normalization by per-element coefficient (EN), fitted by (a) Ordinary Least Squares (OLS) regression and (b) Ridge Regression (RR). OLS is sensitive to outliers and fails to find the optimal coefficients that can effectively narrow the target values.

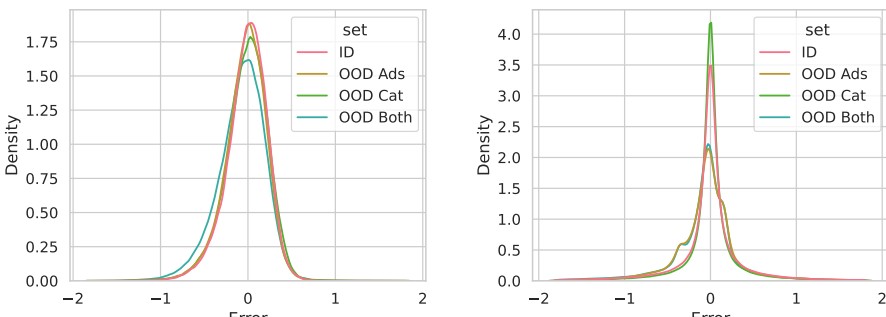

Figure S2: Error distribution using (a) indirect and (b) direct approach. The indirect approach maintains similar error distributions across different splits, indicating its great generalization ability.

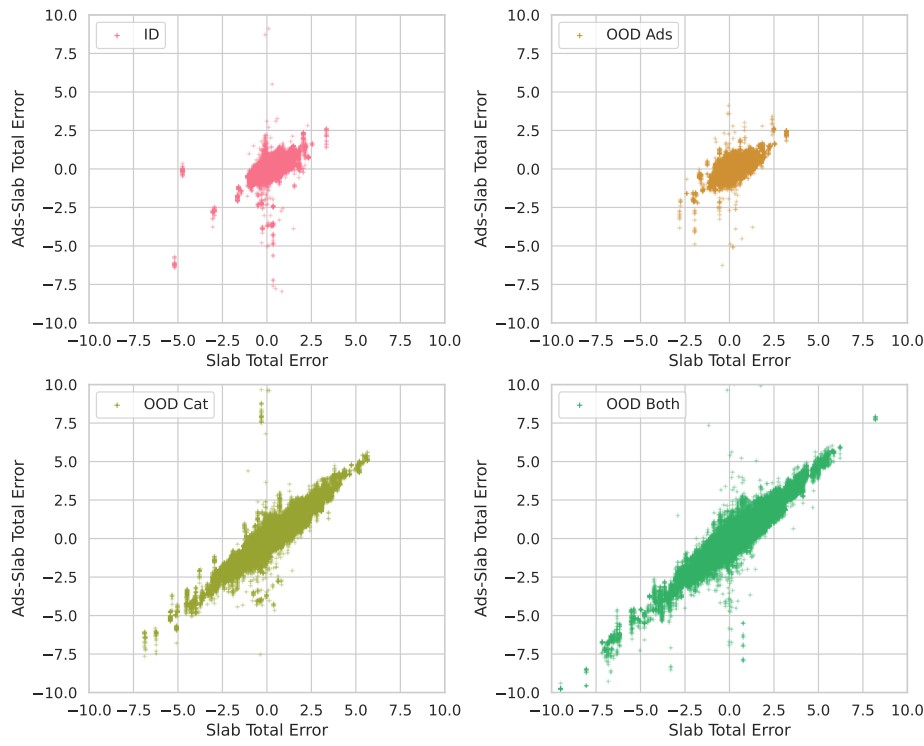

Figure S3: Adsorbate-slab total energy error against slab total energy error for (a) Val ID, (b) Val OOD Ads, (c) Val OOD Cat and (d) Val OOD Both. There is a strong correlation between the errors of the adsorbate-slab and slab for both OOD Cat and OOD Both, which explains why the total energy prediction errors for the adsorbate-slab and slab are high. However, these errors can cancel each other out when used to infer adsorption energy via the indirect approach.

