# OpenReview forum: "Rethinking end-to-end prediction of adsorption energies from a causal perspective"
_ICLR.cc/2025/Conference — ICLR 2025 Conference Withdrawn Submission_

### Official Review · Reviewer_w72e · 2024-11-01

**Soundness:** 3
**Presentation:** 2
**Contribution:** 3
**Rating:** 3
**Confidence:** 3

**Summary:**

This work applies causal reasoning to the prediction of adsorption energy in catalyst systems, with significant improvements to a well-known leaderboard in the area. In particular, identifying that the property of interest is itself a linear combination of other quantities allows the effect of the underlying contributions to be studied. This is nicely motivated with a discussion of potential sources of error. The authors also present analyses to understand what additional data would be helpful to further improve these methods.

**Strengths:**

* The paper has a unique take on prediction tasks for a well-known community challenge and leaderboard for which training/scaling GNNs has been the primary method of improvement by the community.
* The method appears to have state-of-the-art performance on some OC20 validation sets. In particular, validation results of 0.11 eV for S2EF are quite impressive, though I have some questions about the precise training/inference process (see below). If the numbers hold, then this is very impressive and a significant step forward.
* The authors are admirable for their willingness to dive into the details of the underlying dataset and discuss various edge cases and improvements

**Weaknesses:**

1. The paper is written narrowly focused for the catalysis (and specifically open catalyst) domain, which might limit the potential audience. See question below!
2. It is unclear to me whether this was strictly an application of causal modeling to this domain, or if there were methodological improvements required here. If any insights would be helpful to the methodology community, that could be highlighted.
3. The paper would benefit from additional clarify and code/pseudocode for the method implemented.

**Questions:**

1. I am a little bit confused by the overall pipeline for generating the predictions in table 1/2, and think posting code (or, at least, pseudocode in the SI) for the training/inference process would be very helpful. In particular, it’s unclear to me exactly how the various predictions for the slab/adslab energies are incorporated into the final predictions, and what the inference process looks like.

2. When plotting total energy quantities, it might be more helpful to plot energy per atom rather than the total energy

3. A recent article discussed some of the difficulties in predicting total energies in OC20 when only the adsorption energies are converged, and some of the bare slab energies are available (Abdelmaqsoud, Kitchin et al. Cat. Sci. Tech 2024). In particular, I am interested in whether the differences here are mostly related to obtaining a signal on whether the slab reconstructed or not. Similarly, an analysis showing predictions for cases where the slab is significantly different from the initial slab would be very interesting. Or, perhaps analyze residuals for the flagged structures in that paper.

4. Re potential audience weakness above - are there any opportunities to generalize the learnings or approach here to other areas that have energy differences? For example, in inorganic materials, analyses of material energy relative to the hull of known materials may benefit from similar reasoning (see Bartel, Jain et al, npj Comp Materials 2020 and Riebesall, Persson et al. Arxiv 2024). Much of organic chemistry also relies on predictions of energy differences or reaction properties.

5. The authors mention difficulties with out of domain generalization for new adsorbates, and make a statement “Since Egas-ads is typically a constant, it can be disregarded, leaving Eads-slab and Eslab as the primary variables to be predicted”. Egas-ads is in fact linearly related to the composition of the adsorbate, so it is unclear to me if the authors are implicitly fitting this in their process, or if they are simply using the (known) gas phase energies in their final adsorption energy prediction. Either way, the authors might want to consider whether their analysis becomes easier if they subtract that number from the energy of the ads/slab system (so E_ads = (E_adslab-E_gas) - E_slab) ).

---

### Official Review · Reviewer_AVMT · 2024-11-04

**Soundness:** 2
**Presentation:** 2
**Contribution:** 2
**Rating:** 3
**Confidence:** 3

**Summary:**

The paper is focused on improving adsorption energy predictions on the Open Catalyst 2020 dataset. The main contributions of the paper are computing adsorption energy via total energy predictions instead of directly predicting the DFT adsorption energy and using a different linear referencing fitting method.

**Strengths:**

- The application of ML to catalysis is interesting and timely
- Adsorption energy is an important property for catalysis and pushing performance closer to chemical accuracy is an important goal
- Total energy predictions has been shown to work well in other chemical domains such as materials

**Weaknesses:**

- While I do find the ideas presented in the paper interesting, they might be better suited for a chemistry journal.
- The rationale for the casual graph in Figure 1 is a bit difficult to understand, while the slab, ads-slab, and adsorbate energies may not be experimentally observable, they can be calculated with DFT and DFT is the ground truth for the Open Catalyst 2020 dataset.
- It is not clear in the paper that for the “direct” method the adsorption energy label is actually computed using E_ad = E_sys – E_slab – E_gas from DFT values i.e. while the ML model is predicting E_ad directly the label was computed using the various DFT values. It would be helpful to more clearly lay out direct vs indirect approaches including the underlying DFT calculations.
- A new citation to consider/include is Abdelmaqsoud et al. (https://pubs.rsc.org/en/content/articlepdf/2024/cy/d4cy00615a)
- OC22 used a different level of DFT theory than OC20 so using those linear references for OC20 will not work (row 5 in Table 2).

**Questions:**

- In table 1 it mentions that “this table includes only the samples for which we could identify the DFT slab structure from the OC20 validation set” can you further explain what was done here?
- Is table 2 on the test or validation set? If it is the same validation set as table 1, EN-EqV2 performs well on ID and OOD Ads the subsets that were highly filtered compared to OOD Cat and OOD Both.

---

### Official Review · Reviewer_GLZA · 2024-11-04

**Soundness:** 2
**Presentation:** 2
**Contribution:** 1
**Rating:** 3
**Confidence:** 3

**Summary:**

This paper re-examines adsorption energy prediction, proposing an indirect approach based on total energy predictions rather than direct end-to-end methods. Analyzing the causal graph of adsorption energy, the authors argue the indirect approach improves identifiability and generalization. They introduce a regularized graph property normalization for total energy prediction, comparing per-element, per-group, and per-atom methods. Experiments on the OC20 S2EF dataset compare direct and indirect approaches using EquiformerV2, eSCN, SCN, and GemNet-OC architectures. Results are presented for in-domain and various out-of-domain scenarios. The paper also includes ablation studies on graph normalization methods and their impact on total energy and adsorption energy prediction accuracy.

**Strengths:**

This paper presents a perspective on adsorption energy prediction by challenging the direct end-to-end approach and advocating for an indirect method based on total energy prediction. The evaluation on the OC20 S2EF dataset, including diverse in-domain and out-of-domain scenarios, provides an assessment of the proposed approach. The inclusion of ablation studies strengthens the analysis by dissecting the contributions of individual components. However, given its specific focus on catalysis and adsorption energy, the paper might be more appropriate for a specialized journal in catalysis or surface science.

**Weaknesses:**

There is related work of GAMENet by Pablo-Garcia et al. (2023) that should be discussed. More background discussion is required for the introduction. There is a lot of related work, however, terms, jargon, and domain-specific knowledge implied in the paper is not appropriate for this venue. Grammar issues also need to be corrected.

Figure 1 and the discussion of causal perspective does not seem to be necessary. Ultimately, E_ads = E_ads-slab - E_slab (ignoring gas phase), so the models predict the other energies. The errors in the two cancel out, and produce lower MAE. I do not see the connection to causal machine learning, which seems to be the centrepiece of this work, and is also in the title. For example, how does this work connect to Liu et al. (2024)?

The novelty of this work seems to be the comparison of "direct" and "in-direct" predictions. This seems to be an interesting result for a computational chemist in catalysis. However, the models used (EqV2/SCN) are not novel. The authors present new GN, and UN normalization, and claim that it is better than EN, but it seems that Table 2 shows that EN, work from Tran et al. (2023), is better, including for OOD. The work is better suited to a chemistry/catalysis journal than to ICLR.

**Questions:**

Discussed in previous section.

---

### Official Review · Reviewer_ANB7 · 2024-11-04

**Soundness:** 3
**Presentation:** 3
**Contribution:** 2
**Rating:** 3
**Confidence:** 5

**Summary:**

This work investigates the accuracy of machine learning models in predicting adsorption energy, a crucial factor in catalyst discovery. The authors argue that the traditional "end-to-end" approach, which directly predicts adsorption energy from the structure of a catalyst-adsorbate system, is less accurate than an "indirect" approach that infers adsorption energy from the predicted total energies of the catalyst and adsorbate-catalyst system.

**Strengths:**

● This work shows that the indirect approach performs better than direct across different splits (ID and OOD).

● The per-element normalization method significantly improves the accuracy of total energy prediction, achieving halved MAEs compared to direct end-to-end adsorption energy prediction.

● The research provides a theoretical explanation for the error cancellation observed in the indirect approach for out-of-domain catalyst scenarios. The error primarily comes from predicting the energy of unseen catalyst atoms, but this error is canceled out when used to infer adsorption energy.

● Different energy normalization strategies and their impact is measured.

**Weaknesses:**

The major concern I have with this work is the lack of novelty. Most of the conclusions presented in the work have been presented previously in different ways. Only new contribution is different normalization techniques which is marginal (both in terms of contribution and performance).

1. Section S-7 of OC22 paper [1] talks about alternative reference scheme and explores per-atom normalization.
2. Section S-9 of the same work talks about total energy models.
3. This work [2] demonstrates cancellation of energy between the models
4. This work [3] investigates the errors in OC20 again and comes up with the same conclusion on the importance of total energy model and indirect approach

References:

[1] Tran, Richard, et al. "The Open Catalyst 2022 (OC22) dataset and challenges for oxide electrocatalysts." ACS Catalysis 13.5 (2023): 3066-3084.
[2] Ock, Janghoon, et al. "Beyond independent error assumptions in large GNN atomistic models." The Journal of Chemical Physics 158.21 (2023).
[3] Abdelmaqsoud, Kareem, et al. "Investigating the error imbalance of large-scale machine learning potentials in catalysis." Catalysis Science & Technology 14.20 (2024): 5899-5908.

**Questions:**

- There are many points that are not considered in the discussion. Direct models for OC20 technically require more data (or DFT calculations) as it needs slab relaxations for referencing which indirect model doesn't but it has higher inference cost. Which method is the best considering training and/or inference cost tradeoffs?

---

### Note · Authors · 2024-11-15

I have read and agree with the venue's withdrawal policy on behalf of myself and my co-authors.